# Unusual layer-by-layer growth of epitaxial oxide islands during Cu oxidation

Meng Li [1], Matthew T. Curnan [1,2], Michael A. Gresh-Sill[1], Stephen D. House[1,3], Wissam A. Saidi [2✉] & Judith C. Yang[1,3,4✉]

Elucidating metal oxide growth mechanisms is essential for precisely designing and fabricating nanostructured oxides with broad applications in energy and electronics. However, current epitaxial oxide growth methods are based on macroscopic empirical knowledge, lacking fundamental guidance at the nanoscale. Using correlated in situ environmental transmission electron microscopy, statistically-validated quantitative analysis, and density functional theory calculations, we show epitaxial $Cu_2O$ nano-island growth on Cu is layer-by-layer along $Cu_2O(110)$ planes, regardless of substrate orientation, contradicting classical models that predict multi-layer growth parallel to substrate surfaces. Growth kinetics show cubic relationships with time, indicating individual oxide monolayers follow Frank-van der Merwe growth whereas oxide islands follow Stranski-Krastanov growth. Cu sources for island growth transition from step edges to bulk substrates during oxidation, contrasting with classical corrosion theories which assume subsurface sources predominate. Our results resolve alternative epitaxial island growth mechanisms, improving the understanding of oxidation dynamics critical for advanced manufacturing at the nanoscale.

[1] Department of Chemical and Petroleum Engineering, University of Pittsburgh, Pittsburgh, PA, USA. [2] Department of Mechanical Engineering & Materials Science,, University of Pittsburgh, Pittsburgh, PA, USA. [3] Environmental TEM Catalysis Consortium (ECC), University of Pittsburgh, Pittsburgh, PA, USA. [4] Department of Physics and Astronomy, University of Pittsburgh, Pittsburgh, PA, USA. ✉email: alsaidi@pitt.edu; judyyang@pitt.edu

Advanced manufacturing of nanostructured metal oxides (MOs) is essential for myriad applications including energy, electronics, sensors, photocatalysts, bio-medicine, and recently for quantum computing[1–12]. Thus, precise, scalable synthesis and processing of nanostructured MOs are in much demand[13]. Microfabrication methods, such as thermal oxidation, reactive sputtering, and atomic layer deposition, are promising approaches for preparing large batches of epitaxially nanostructured MO[14,15]. However, current nano-oxide fabrication methods, for which oxidation is a vital step, are empirically based. To better predict and control the shape of nanostructured MOs, a fundamental understanding of the oxide nanocrystal growth process is essential. Established oxidation theories of Wagner[16] and Cabrera–Mott[17] treat the oxidation process exclusively from a macroscopic viewpoint assuming simplified, continuous uniform layers. While these models have been successful in guiding fabrication of amorphous oxide films—such as $SiO_2$—and corrosion mitigation, they have little predictive power for describing nanostructured MOs due to their lack of atomic crystalline considerations. This shortcoming greatly hinders the industrial manufacturing of nanostructured MOs.

Borrowing from thin-film growth theories, MO nanocrystals that attach to a metal substrate epitaxially during the growth of three-dimensional (3D) oxide islands are explained using the Stranski–Krastanov (layer-plus-island) growth mode[14,18]. However, this model is defined from an interfacial energy viewpoint, leaving the kinetic process of how these nanostructures form uncertain. One well-accepted kinetic process is the multilayer growth mechanism, which explains the formation of 3D islands as the simultaneous growth of multiple layers stacked parallel to the substrate surface, forming "wedding cake"-shaped islands[19–21]. For example, islands on (100) substrates form by the concurrent growth of multiple stacked layers along the (100) plane[21]. However, 3D islands with faceted crystal surfaces are also widely observed, such as pyramidal Ge and Si islands in quantum dots[22–24], nano-wedge-shaped Fe islands[25], and 3D $Cu_2O$ islands on Cu[26–29]. While these faceted crystal surfaces are at variance with the multilayer growth mechanism, it remains unclear whether the deviations are due to the early or the later stages of the island growth. The lack of direct observation of the growth dynamics at the atomic scale has hindered establishing a fundamental mechanistic explanation for the growth of the 3D epitaxial islands. Recent developments in in situ environmental transmission electron microscopy (ETEM)—with which material systems can be examined under relevant reaction conditions—offer a solution to this problem, enabling the direct observation of growth dynamics[4,30,31]. However, the results heretofore have been qualitative at best. Extracting statistically meaningful quantitative atomic-scale growth kinetics from the in situ movies, which is critical for understanding atomic-scale growth mechanisms, has become the new challenge.

Herein we perform in situ ETEM oxidation experiments on copper—the most well-studied model material for oxidation that forms epitaxial oxide islands—to provide direct, atomic-scale observations of the growth dynamics of 3D epitaxial oxide islands during oxidation. Quantitative atomic-scale information was extracted using advanced image analysis techniques. By correlating the experimental observations and statistical validation of growth kinetics with density functional theory (DFT) modeling, we present an unusual epitaxial layer-by-layer growth mechanism for the oxide island along a preferred surface facet, unforeseen by previous crystal growth theories.

## Results

**Layer-by-layer $Cu_2O$ growth along $Cu_2O(110)$.** 3D $Cu_2O$ islands were formed by oxidizing single-crystalline Cu films inside the ETEM at 300 °C under 0.3 Pa $O_2$. In agreement with previous studies[32,33], these $Cu_2O$ islands share cube-on-cube epitaxy with the Cu substrate. The $Cu_2O$ islands on Cu(100) were reported to follow the Stranski–Krastanov (S–K) growth mode, in which a transition from 2D wetting layers to 3D islands was observed beyond a critical thickness[34]. According to previous models[33,35], the oxide is expected to grow along the Cu surface, such as along $Cu_2O(100)$ on Cu(100). However, as shown in Movie S1 and Fig. 1, we found that the $Cu_2O$ islands on both Cu(100) and (110) surfaces (Supplementary Note 1, Supplementary Figs. 2–5, and Supplementary Movies 2–4) grew along the $Cu_2O(110)$ planes in a layer-by-layer adatom growth mode. This is usually observed in Frank–van der Merwe (F–M) growth where the interface mismatch energy is negligible, leading to the formation of a thin-film, instead of islands, along the substrate surface. Our study shows that although the resultant $Cu_2O$ islands follow the S–K growth mode, the formation of each 3D island follows a layer-by-layer growth along a certain plane that is not necessarily parallel to the substrate surface, contradicting classical predictions.

**$Cu_2O$ monolayer growth kinetics.** To better understand the growth kinetics of each monolayer, we performed quantitative analysis on the boxed area in Fig. 1d on the atomically aligned movie to measure the size evolution of each layer over time (Fig. 1f, Supplementary Note 3). As illustrated in Fig. 1e, each new layer is a 2D flake that grows in both planar directions. Although TEM images only provide 2D through-thickness projections of the growing new layer, given that the two directions are equivalent {110} planes, the projected length evolution in one dimension can be used to estimate the growth in both directions. The nucleation sites of each $Cu_2O$ monolayer (Fig. 1g and Supplementary Fig. S6) were randomly distributed on the previously grown layer, indicating identical nucleation energetic favorability across all sites. Despite that the layer edges typically proceeds in stairs (Fig. 1g), the total projected lengths ($l$) all exhibited a similar smooth growth trend following a cube root relationship with time ($t$): $l^3 = At$ (referred to as cubic relationship for short hereafter, Fig. 1h, Supplementary Fig. 14, and Supplementary Note 3). As a result, the growth rate of the island size is quasi-linear along the [110] direction and cubic along [100] and [010] directions. Interestingly, typical oxide thickness curves found in bulk Cu oxidation experiments in this temperature range also follow cubic rates[36], which were explained based on classical oxidation theory[17] due to the formation and diffusion of cation vacancies through an oxide layer that fully covers the metal surface. However, this explanation does not apply to our sample, since the metal surface near the oxide island is still exposed. Instead, we argue that the cubic growth rate of each oxide monolayer could be explained using the diffusion-limited 2D growth kinetics of F–M thin-film growth[19]. In this growth mode, a full layer forms by coalescence of several single-monolayer-thick 2D flakes, where each flake grows with $l^3 \sim t$ scaling[37–39]. The main difference with the thin-film F–M growth is that the observed growth of the oxide monolayer in our samples is due to the nucleation of a single "flake", rather than many flakes as in thin-films. This difference is likely because the effective substrate for the growth of the oxide layer is relatively small compared to typical thin-film substrates. Presumably, as the MO islands grow in size, layer formation through the coalescence of more than one flake would occur. Hence, the observed $l^3 \sim t$ growth kinetics of each monolayer indicates a diffusion-limited process, possibly due to surface diffusion of Cu and O atoms to form adatoms at the $Cu_2O$ monolayer edge.

The growth trajectories also exhibited coordinated increments and oscillations between multiple layers (Fig. 1g–i). To substantiate

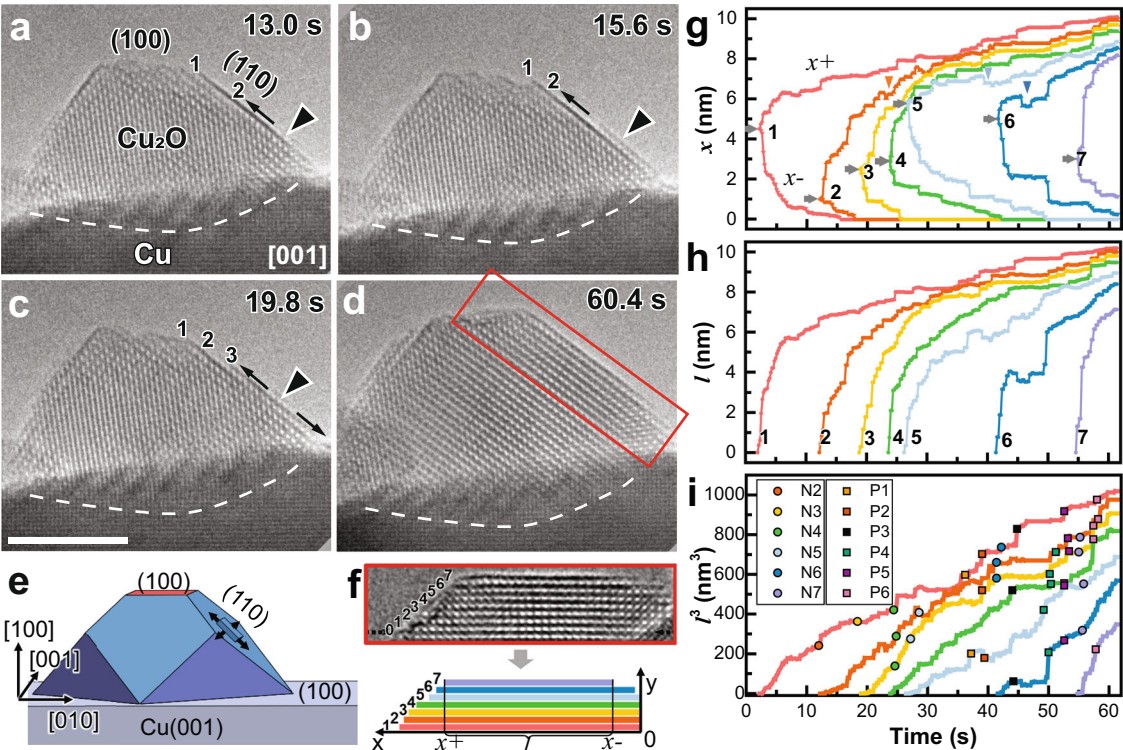

**Fig. 1 Layer-by-layer growth of Cu₂O island along Cu₂O(110) during Cu(100) oxidation. a–c** Snapshots from Supplementary Movie 1 showing adatom growth of the 2ⁿᵈ Cu₂O new layer at 300 °C and $p_{O2} = 0.3$ Pa. The layer nucleation site (triangle) and growth direction (arrow) are indicated. Scale bar: 5 nm. **d** The island top forms a flat Cu₂O(100) plane over time. **e** Schematic 3D model of the Cu₂O island with a growing new layer. **f** The boxed area from **d**, reoriented for growth trajectory measurement, with a corresponding schematic defining the measured data plotted in **g–i. g** Growth trajectory coordinates of the left (x+) and right (x−) ends of each layer with time, namely when measuring from the right side of the image defined in **f**. Nucleation sites on each layer, marked by gray arrows, indicate a random site distribution. These two ends show stepwise growth with oscillations marked by triangles of matching colors. **h** The projected length (l) of each layer shows a similar trend with smoother curves. **i** Statistically defined breakpoints in growth rates indicate nucleation events (N) and interlayer atom diffusion events (P).

these sudden changes, a multivariate time-series statistical analysis[40] was performed ($l^3$ vs. $t$) to evaluate when breaks in otherwise continuous growth occur[37,38,41]. The statistically defined structural breaks[40], shown in Fig. 1i and Supplementary Note 4, are attributed to two types of events, namely the nucleation of new layers (N2–N7) and concerted diffusion events (P1–P6) describing the simultaneous change of growth rates among several layers. Nucleation events generally led to a growth rate decrease in previously grown layers, indicating that Cu and O attachment to the nucleating new layer is preferred over attachment to previous layers. This is likely caused by the Ehrlich–Schwöbel effect, in which downward diffusion across a surface step is prohibited due to an extra energy barrier[42–44]. Concerted diffusion events generally showed sudden decreases in the growth rates of new layers and increases in those of previous layers. This indicates a cross-layer diffusion of Cu/O sourced from the new Cu₂O monolayer to feed the growth of former layers, which corresponds to adjustment of the top of the Cu₂O island from an initially zig-zagged surface to a flat Cu₂O(100) facet (Supplementary Figs. 7–8). Hence, both the oxide monolayer growth trend and variations in monolayer growth indicate a diffusion-limited layer-by-layer growth process resembling the F–M thin-film growth mode, although the overall oxide island follows the S–K mode.

**DFT calculations on Cu₂O monolayer growth mechanism.** To better understand why the oxide grows along the Cu₂O(110) plane in disagreement with classical theories, DFT calculations were

performed to study the earlier-stage atom-by-atom oxide growth events. Gas–solid interfacial energies ($\gamma$) were calculated for the (100) and (110) Cu₂O surface planes that are predominantly observed in our experiments. As shown in Fig. 2a and detailed in Supplementary Note 5, CuₓOᵧ surface units were sequentially added to these planes to simulate layer growth. For flat Cu₂O surfaces (column *i* in Fig. 2a), Cu–O terminated Cu₂O(110) had the lowest $\gamma$. Upon adding CuₓOᵧ surface units, Cu₂O(110) surfaces with exposed Cu–O layers invariably had the lowest $\gamma$. These surfaces include structures *i* and *iii* for Cu–O terminated Cu₂O(110) and structure *ii* for Cu-terminated Cu₂O(110), and their favorability is attributable to their terminal, ionically bonded O–Cu–O chains (Supplementary Figs. 21–23). In contrast, O-terminated Cu₂O(100) had the highest $\gamma$, regardless of the number of CuₓOᵧ surface units added to it. Such instability coincides with the undercoordination of exposed O atoms (Supplementary Fig. 20). Since Cu₂O(100) structures must produce less-stable O terminations during its growth, oxide growth along Cu₂O(110) is preferable to Cu₂O(100). Simulated adatom adsorption events forming CuₓOᵧ surface unit *ii* (hollow data points in Fig. 2a and Supplementary Note 6) further support this conclusion. Therefore, $\gamma$ trend comparisons show that Cu₂O(110) forms thermodynamically more favorable flat surfaces, grown Cu₂O monolayers, and single adatom interfaces than Cu₂O(100), in agreement with the experimental results.

Nudged elastic band simulations of the diffusion events along the oxide island surface further verified the experimentally observed layer-by-layer growth kinetics along the Cu₂O(110) plane (Supplementary Note 7). Figure 2b compares the preferred

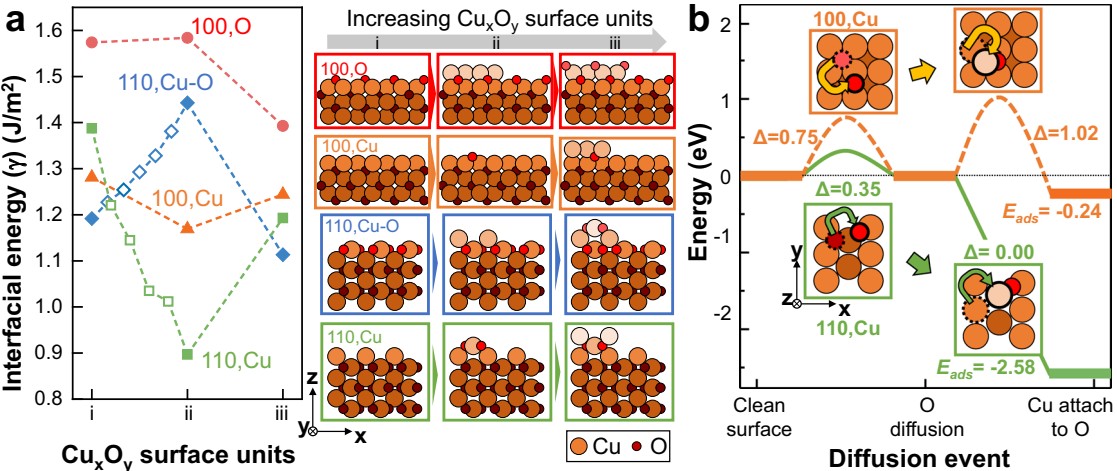

**Fig. 2 DFT-calculated interfacial energies and diffusion energies for Cu₂O (110) and (100) surfaces. a** Most favorable structure $\gamma$ during Cu₂O monolayer growth for Cu₂O(110) with Cu (green) and Cu–O (blue) terminations, and Cu₂O(100) with Cu (orange) and O (red) terminations. The corresponding atomic structures are shown to the right of the plot, with matching border colors. Hollow data points between *i–ii* show adatom adsorption events forming the first $Cu_xO_y$ surface unit. **b** Most-favorable Cu and O diffusion events on Cu₂O surfaces linking diffusion barrier ($\Delta$) and adsorption energy ($E_{ads}$). Insets with matching border colors show top–down views of the corresponding atomic structures. Cu and O atoms are colored in orange and red, respectively. Brighter/darker colors indicate a higher/lower $z$ position.

diffusion mechanisms required to make an oxide layer on Cu terminated Cu₂O(110) and Cu₂O(100) surfaces. The most favorable rate-limiting diffusion process for forming Cu–O layers on Cu-terminated Cu₂O(110) is single O diffusion with inter-channel endpoints (diffusion barrier of $\Delta = 0.35$ eV), and the matching Cu diffusion process to place Cu on top of that O is barrierless with favorable adsorption energy ($E_{ads} = -2.58$ eV). In comparison, the most favorable rate-limiting O diffusion process on a Cu-terminated Cu₂O(100) surface is in-channel single O diffusion ($\Delta = 0.75$ eV). The matching Cu diffusion process places Cu on top of that O, yielding a large barrier ($\Delta = 1.02$ eV) and less-favorable adsorption energy ($E_{ads} = -0.24$ eV). The corresponding process for forming Cu layers on Cu–O terminated Cu₂O(110) is in-channel single Cu diffusion ($\Delta = 0.42$ eV, Supplementary Fig. 29). Rate-limiting step comparisons favor Cu₂O(110) over Cu₂O(100) not only across the initial and final states of the oxide layer formation process (0.42 eV vs. 1.02 eV) but also for each transient diffusing atom composition. Therefore, Cu and O prefer to diffuse to Cu₂O(110) island growth fronts over Cu₂O(100) fronts, further validating prior experimental outcomes.

**Cu source for Cu₂O island growth.** The source of Cu during Cu₂O growth also warranted investigation. Traditional oxidation theory[17] argues that Cu is supplied from the metal‖oxide interface through diffusion across the oxide, leading to an interface shift toward the metal side. However, as seen in Figs. 1 and Supplementary Fig. 6, the Cu‖Cu₂O interfaces predominantly remained unchanged during the oxidation process, particularly during the initial period of oxidation when there are few nucleated Cu₂O islands. This indicates that there must be other sources for Cu instead of the substrate Cu. A recent study found Cu₂O island growth with gradual height decreases of the surrounding Cu surface steps[45], inferring that Cu detaching from step edges might be the source. However, direct evidence for this claim has been lacking. Figure 3 and Supplementary Movie 5 show that during oxidation of Cu(100) facets with several one-atomic-layer-high surface steps, these steps retreated when Cu₂O grew, while the Cu‖Cu₂O interface remained unchanged. Due to the

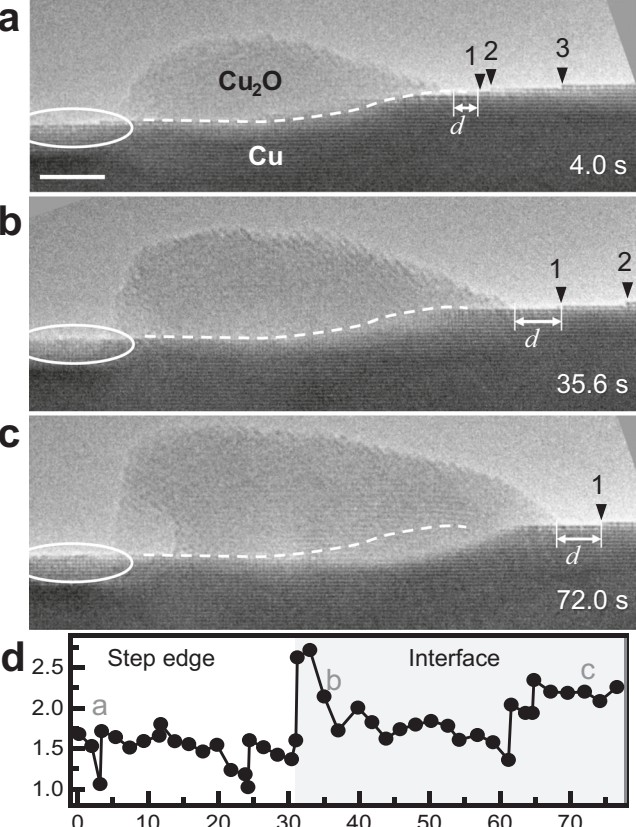

**Fig. 3 Cu sources for Cu₂O nano-island growth. a**, **b** Cu₂O grows while surface steps (1–3, marked by triangles) retreated and the Cu‖Cu₂O interface (dashed lines) remained unchanged. (Scale bar: 5 nm). **c** When surface steps were far from the Cu₂O island, Cu₂O continued to grow and the Cu‖Cu₂O interface started to migrate towards Cu. **d** A plot of the measured distance ($d$) between the Cu₂O island and step edge "1" with time. $d$ suddenly increased at ~31 s, leading to the transition from step edge Cu to bulk Cu consumption.

thickness difference between the Cu film and the Cu$_2$O island, the amount of Cu lost from the surface steps is comparable to the amount of Cu added to Cu$_2$O. This indicates that Cu detaching from step edges is the source of Cu for Cu$_2$O growth in the early oxidation stage. Later, the Cu||Cu$_2$O interface migrates toward the Cu substrate, indicating that in later stage oxidation, the substrate serves as the Cu source. The transition between Cu sources is determined by the distance from the nearest step edge (Fig. 3d), which is explainable by Cu diffusion. When the oxide island is near a step edge, Cu detached from surface steps can easily diffuse to the oxide island via surface diffusion. However, when there are very few surface steps or the oxide island is far away from the surface steps, bulk Cu diffusion from the Cu substrate to the gas||oxide interface becomes more efficient. As shown in Supplementary Note 8, even on reconstructed Cu surfaces, the diffusion barrier of Cu surface diffusion is still lower than that of bulk diffusion, leading to less-faceted oxide shapes during the interfacial Cu sourcing stage.

**Mechanism of 3D Cu$_2$O island growth.** Based on the above discussion and the energetic data summarized in Supplementary Table 7, the mechanism of the unusual epitaxial oxide island growth processes during Cu oxidation, summarized in Fig. 4, is:

a. Due to surface reconstruction, O$_2$ dissociation on the Cu surface is inhibited, so O adatoms are provided by O$_2$ dissociative absorption on Cu$_2$O surfaces[2,46]. The preferred diffusion barriers and adsorption energies of O on Cu$_2$O (110) over those on Cu$_2$O(100) suggest more diffusing O atoms will be present on the Cu$_2$O(110) surface.

b. When there are Cu surface steps nearby, Cu adatoms detached from Cu step edges diffuse to Cu$_2$O surfaces via surface diffusion. Because of the preferred diffusion barriers and adsorption energies of Cu on Cu$_2$O(110), more Cu atoms will diffuse on Cu$_2$O(110) than Cu$_2$O(100). Due to the lower surface energy of Cu$_2$O(110) and more favorable adsorption energies of Cu and O on the Cu$_2$O(110) step, Cu$_2$O nuclei will form on Cu$_2$O(110). This is followed by the growth of Cu$_2$O monolayers in an atomic adsorption process directed toward the growth front of the new layer. The vapor deposition process to grow oxides can be viewed as an extreme case of this scenario, where there are sufficient mobile Cu atoms present to directly react with O atoms[45].

c. The new Cu$_2$O layer grows in this adatom growth method until the edge of the layer reaches the ridge of the previous Cu$_2$O layer, then a new Cu$_2$O layer nucleates following the steps in (a) and (b). This leads to the observed layer-by-layer growth along Cu$_2$O(110). For each monolayer, the growth of a new Cu$_2$O layer on the Cu$_2$O(110) facet follows the diffusion-limited Frank–van der Merwe growth process with cubic growth rate $l^3 \sim t$. However, due to the preference of Cu$_2$O(110) over Cu$_2$O(100) in both kinetics and energetics, the overall Cu$_2$O island follows the Stranski–Krastanov growth model. When the Cu surface steps are far away, substrate Cu will feed Cu$_2$O growth by interfacial diffusion via place exchange with Cu vacancies in Cu$_2$O islands.

## Discussion

Using DFT, we have investigated the energetics of several most probable diffusion paths, and corresponding thermodynamic states, that are proposed to underline studied experimental observations. However, a complete understanding of studied oxide growth dynamics is beyond the capabilities of DFT alone,

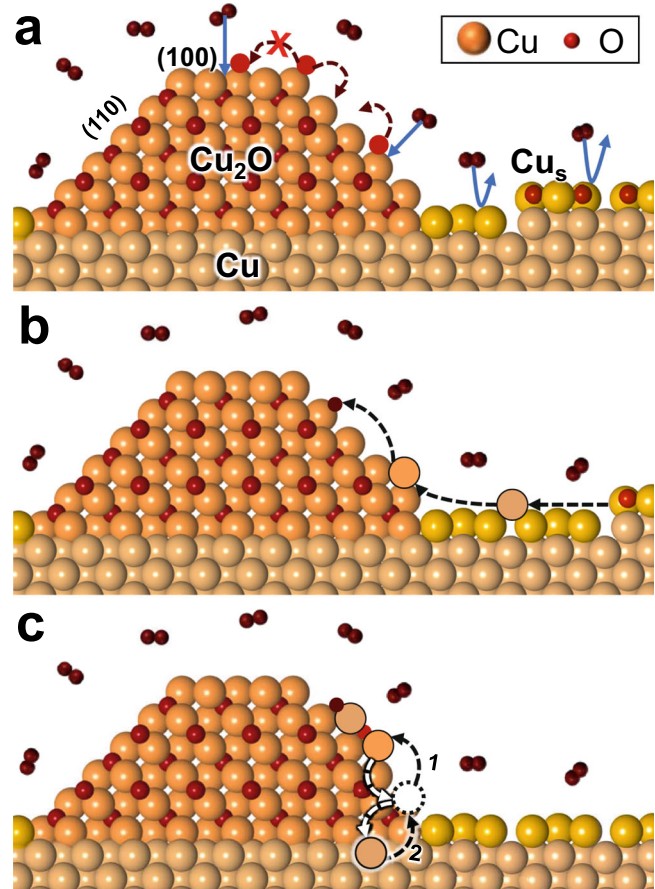

**Fig. 4 Schematic of Cu$_2$O growth mechanism. a** Dissociative adsorption of O$_2$ (red atoms) is blocked on the reconstructed Cu surface (Cu$_s$, gold atoms) and performed on Cu$_2$O. Given lower diffusion barriers and favorable O adsorption energies on Cu$_2$O(110), more O atoms segregate toward Cu$_2$O(110). **b** When Cu steps are nearby, Cu adatoms detached from Cu step edges, diffuse to Cu$_2$O islands, and attach to O atoms to form Cu$_2$O monolayers, given favorable Cu and O $E_{ads}$ on Cu$_2$O(110) steps. **c** When Cu surface steps are far away, substrate Cu will feed the growth of Cu$_2$O via interface diffusion, namely through place exchange with Cu vacancies (dashed circle). Cu atoms from Cu$_2$O, the Cu surface, and the Cu bulk are colored orange, gold, and beige, respectively. O atoms are colored dark red.

especially when considering the many possible diffusion processes that induce concerted oxide nucleation and growth processes over multiple island layers. Comparisons between statistical conclusions and ETEM observations made in "Correlating Statistical Results with Experimental Observations" (Supplementary Note 4) demonstrate that limitations in Cu sourced from adjacent island layers contribute to observed island shapes and relative layer growth rates. Therefore, the combination of surface orientations and terminations predicted by simulation, island shape evolution by ETEM observations, and relative growth rates characterized by statistical conclusions, is needed to completely depict oxide growth dynamics.

Our results provide direct, atomic-scale growth dynamics of 3D epitaxial oxide island growth. Instead of multilayer growth along substrate surfaces to form wedding-cake shaped islands, we found the growth of 3D epitaxial oxide islands follows a layer-by-layer growth mechanism along a preferred facet. The growth kinetics of each oxide monolayer is consistent with predictions from the diffusion-limited 2D Frank–van der Merwe growth model for thin-films[37]. To our knowledge, this is the first atomic-resolution

experimental proof of the atomic-level growth dynamics of 3D islands. Our study sheds new light on the epitaxial oxide growth mechanism and provides a deeper understanding of the dynamic processes involved in initial oxidation, which will ultimately help to precisely predict, design, and control nanostructured oxide growth. Our findings would apply to other metals—such as Al[47], Ni-Cr[4,48], Mo[30], Mg[49,50] and Ag[51]—where a similar layer-by-layer oxide growth was observed for the islands, though without confirmation on the early stages. Moreover, this work demonstrates that with meticulous in situ TEM experiments and advanced data analysis, statistically meaningful quantitative atomic-scale growth kinetics can be resolved. When complemented with correlated theoretical simulations, such work will promote the understanding of nanoscale dynamics to a new level.

## Methods
Provided in Supplementary Information.

## Data availability
All data is available in the main text or the Supplementary Materials (Supplementary Movies S1–S5, Methods, Supplementary Notes 1–8, Supplementary Notes 1–8, Supplementary Figs. 1–31, and Supplementary Tables 1–7).

## Code availability
The code for in situ movie analysis and statistical analysis are available from the corresponding author on request.

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

## Acknowledgements

We acknowledge Mr. Xianhu Sun (SUNY Binghamton), Drs. Hao Chi and Henry Ayoola (University of Pittsburgh) for helpful discussions and assistance in experiments. The experimental work was performed at the Petersen Institute of NanoScience and Engineering (PINSE) Nanoscale Fabrication and Characterization Facility (NFCF) at University of Pittsburgh. We thank NFCF staff Mr. Matt France and Dr. Susheng Tan for their assistance. This research used resources of the Environmental TEM Catalysis Consortium (ECC), which is supported by the University of Pittsburgh and Hitachi High Technologies. Computational resources were provided by the University of Pittsburgh Center for Research Computing (CRC), the Extreme Science and Engineering Discovery Environment (XSEDE) supported by the National Science Foundation (NSF OCI-1053575), and at the Argonne Leadership Computing Facility, which is a DOE Office of Science User Facility supported under Contract DE-AC02-06CH11357.

## Author contributions

J.C.Y. and W.A.S. conceived and directed the project. M.L. conducted the experiments, data analysis, and drafted the manuscript. M.T.C. conducted D.F.T. simulations and statistical analysis. M.A.G. and S.D.H. assisted with the data analysis. All authors contributed to results discussion and manuscript refinement.

## Funding

This work is supported by the National Science Foundation (NSF DMR-1410055, NSF DMR-1508417, and NSF CMMI-1905647).

## Competing interests

The authors declare no competing interests.
