## [Peer Review File · Nature Communications]

REVIEWER COMMENTS

Reviewer #1 (Remarks to the Author):

This is an excellent study into the growth of oxides on metal surfaces. I agree with the authors that this topic requires more in-depth detailed studies at the atomic level. Often the growth mode is defined from an interfacial energy viewpoint, leaving the kinetic process of how these nanostructures form uncertain. In my view this study makes a substantial contribution into this field using the system of CuO₂ on the surface of Cu(001) as a model system. The study is based on excellent experimental data obtained using in situ environmental transmission electron microscopy, a state of the art instrument in high resolution electron microscopy. The experimental study is supported by the DFT calculations. There is an excellent analysis section bringing the experiment and theory together into a clear unified picture of the process in this model oxide system. The key aspects of the oxide growth in this system are considered, including the dissociation of oxygen, migration of cations towards the sites of forming the oxide, direction of growth of the oxide.

The conclusions are clear and well presented.

The most important conclusion is that it is clearly demonstrated that instead of multilayer growth along substrate surfaces to form wedding-cake shaped islands, the authors found the growth of 3D epitaxial oxide islands follows a layer-by-layer growth mechanism along a preferred facet. The growth dynamics of each oxide monolayer is consistent with predictions from the Frank-van der Merwe growth model for thin films in the diffusion limit.

These results are highly relevant in the science and technology of growth of oxides and even other binary materials.

The paper is well written.

I recommend that this paper is published in Nature Communications.

My only comment is that the captioning and explanation of figure 1(g) could be clearer. A clearer indication of the point of nucleation, -x and +x, the coloured arrows and an explanation of what the grey arrows are indicating would be beneficial.

Prof. Igor Shvets
Chair of Applied Physics
Trinity College
University of Dublin
Dublin 2
Ireland

Reviewer #2 (Remarks to the Author):

The authors are to be complemented on their detailed study of Cu oxidation using ETEM and extensive modelling providing added insight into this phenomenon. Their challenging environmental TEM experiments and modelling results are very impressive, and are of special interest in oxidation research, particularly in terms of the initial stages of oxidation of Cu. However, this research is more appropriate for other more specialist journals (e.g., physical chemistry, oxidation). This research definitely merits publication, but it is not best suited for Nature Communications but should be published in a more focused journal.

Reviewer #3 (Remarks to the Author):

This is an interesting study of an important topic that I hope will be published in a revised form. I believe that major revisions to the current manuscript are required and give details below.

It is certainly true, as stated by the authors, that elucidating atomic mechanisms for oxide surface growth at surfaces is of great interest. The measurements and calculations presented may well make a contribution to understanding this for Cu₂O growth on Cu. As stated below it is hard to be sure given the presentation of the data in this article. It is not, however, established clearly that the results “provide an approach to study growth dynamics of manufacturing processes” as claimed in the Abstract. Neither environmental TEM or DFT are new approaches nor is their application to the growth of oxides at surfaces. In this regard a complete article must provide at least a brief review of the previous use of ETEM, its use for similar problems and suitable citations. This is missing from the current work. For these reasons this claim should be modified or justified and a brief review provided.

The presentation of the data is couched in a description of the science that makes it very difficult to determine what the authors consider to be novel in this work and which of their conclusions are supported by the data presented.

The main results of the article are i) nano islands formation growing on 110 planes ii) thickness depending on the cube root of time – hence diffusion limited growth iii) a transition from step edges to substrate for the source of Cu.

As the authors note, growth of Cu₂O by nano-island formation has been observed previously. Refs 25-28 in the current article for example. The authors seem to claim that the main import of this observation is that it “contradicts classical models”. As the classical models say nothing about the habit of the particles in the growth layer it isn’t clear what is meant. It is also stated that the “cubic relationship in time”, in fact a cube root relationship, provides evidence of a diffusion limited growth mode that is (and here the final conclusion is unclear to me) somehow intermediate between the Stranski-Krastanov and the Frank-van der Merwe (FvdM) models. As a main conclusion of the article this is vague.

The dependence of layer thickness on time and its relationship to the atomistic structure of the particles is potentially very interesting. The detailed discussion of this in terms of the computed DFT energetics is interesting but unconvincing. Certainly the claim (in part c) of this discussion) that the layer by layer growth of a nano-facet has a cube root time dependence because it resembles multilayer FvdM diffusion limited growth of a film is not obvious and needs to be explained more carefully. This raises the question about the role of the DFT calculations in this article. These are presented as a comprehensive examination of the atomistic growth mechanism. In reality the surface energies of facets are computed (these are available in the literature using very similar methods) and these are supplemented by the attachment of some CuxOy species and diffusion pathways for a few surface barriers. I accept that this is a very complicated process and its full elucidation in DFT is beyond current capabilities. For this reason a more balanced presentation of the results including the caveats about their application would be more convincing. If the authors believe that the surface energies, attachment mechanisms and diffusion are quantitative and in some sense representative of the overall growth they should be able to combine them to produce a

kinetic growth model that correctly predicts both shape and growth rate – is that possible ?

The transition of the Cu source from step edges to substrate is an interesting result and is convincing. The videos of the oxide growth are also very interesting and a very nice example of what can be achieved with ETEM.

Relatively minor point additional points are:

It is best to avoid undefined mnemonics in the Abstract

There are numerous minor errors in the grammar. These make reading the article difficult and in places obscure the meaning. I found myself rescanning sentences multiple times to extract the meaning. Proof reading or use of an accurate automatic tool should enable these to be removed and for both the clarity and readability of the article to be improved.

Point-by-point Response to Referee's Comments

Note: The reviewers' comments are in blue and italic, our responses are in black, and the changes made to the manuscript are highlighted in bold red.

Reviewer #1 (Remarks to the Author):

This is an excellent study into the growth of oxides on metal surfaces. I agree with the authors that this topic requires more in-depth detailed studies at the atomic level. Often the growth mode is defined from an interfacial energy viewpoint, leaving the kinetic process of how these nanostructures form uncertain. In my view this study makes a substantial contribution into this field using the system of CuO₂ on the surface of Cu(001) as a model system. The study is based on excellent experimental data obtained using in situ environmental transmission electron microscopy, a state of the art instrument in high resolution electron microscopy. The experimental study is supported by the DFT calculations. There is an excellent analysis section bringing the experiment and theory together into a clear unified picture of the process in this model oxide system.

The key aspects of the oxide growth in this system are considered, including the dissociation of oxygen, migration of cations towards the sites of forming the oxide, direction of growth of the oxide.

The conclusions are clear and well presented.

The most important conclusion is that it is clearly demonstrated that instead of multilayer growth along substrate surfaces to form wedding-cake shaped islands, the authors found the growth of 3D epitaxial oxide islands follows a layer-by-layer growth mechanism along a preferred facet. The growth dynamics of each oxide monolayer is consistent with predictions from the Frank-van der Merwe growth model for thin films in the diffusion limit.

These results are highly relevant in the science and technology of growth of oxides and even other binary materials.

The paper is well written.

I recommend that this paper is published in Nature Communications.

My only comment is that the captioning and explanation of figure 1(g) could be clearer. A clearer indication of the point of nucleation, -x and +x, the coloured arrows and an explanation of what the grey arrows are indicating would be beneficial.

*Prof. Igor Shvets
Chair of Applied Physics
Trinity College
University of Dublin
Dublin 2
Ireland*

Reply: We would like to thank the reviewer for the nice summary and comments on our paper.

We have modified the caption of Figure 1(g) as follows:

“... (g) Growth trajectory coordinates of the left (x_+) and right (x_-) ends of each layer with time, namely when measuring from the right side of the image defined in f). Nucleation sites on each layer, marked by gray arrows, indicate a random site distribution. These two ends show stepwise growth with oscillations marked by triangles with matching colors. (h) The projected length (l) of each layer shows a similar trend with smoother curves.”

Reviewer #2 (Remarks to the Author):

The authors are to be complemented on their detailed study of Cu oxidation using ETEM and extensive modelling providing added insight into this phenomenon. Their challenging environmental TEM experiments and modelling results are very impressive, and are of special interest in oxidation research, particularly in terms of the initial stages of oxidation of Cu. However, this research is more appropriate for other more specialist journals (e.g., physical chemistry, oxidation). This research definitely merits publication, but it is not best suited for Nature Communications but should be published in a more focused journal.

Reply: We would like to thank the reviewer for the summary and comments on our paper. Although the experiment is focused on the oxidation of Cu, the atomic-scale growth dynamics we presented are vital to understanding nanostructural growth, which is important to many other applications such as quantum materials, catalysts, and electronics.

Here, we summarize the motivation and significance of our work and discuss breakthroughs in our technical and scientific innovations:

Motivation and general interest: Nanostructured metal oxide has broad applications in energy, catalysts, electronics, sensors, bio-medicine, and so on. However, current methods for epitaxial nanostructured oxides growth are still empirical due to a lack of fundamental understanding of their growth mechanism. This has become a bottleneck for precise, scalable fabrication of nanostructured metal oxides for their industrial applications. Quantitative atomic-scale experimental data on epitaxial oxide growth dynamics, in correlation with theory, are vital to resolve this problem.

Breakthrough technical innovations:

Automated *in situ* HRTEM data analysis method: Through meticulous *in situ* environmental TEM experiments, we observed the atomic scale dynamic process of epitaxial oxide island growth during Cu oxidation. However, extracting quantitative data from the experimental movies and the ensuing data analysis are another great obstacle to overcome. This is due to the lack of tools that are required to bridge between multiple disciplines, including materials science, microscopy, computer science, and statistics. Through years of efforts, we developed an automated *in situ* HRTEM data analysis method that could extract quantitative data from HRTEM movies with minimum human intervention, saving researchers from arduous manual frame-by-frame processing and analysis and greatly increasing productivity.

Multivariate time series statistical analysis on *in situ* TEM data: Statistical analysis has been widely used in experimental research to help elucidate trends and connections behind the data. However, as far as we know, comparative modeling of simultaneous trends to determine concerted structural breaks across multiple dynamic processes has never been done in the *in situ* TEM community, or even in most physical sciences (excluding climatology, medicine, and a few other fields). Reasons for this could include difficulty in obtaining sufficient data sampling and the potential non-normality of studied trends, both of which are accommodated by our employed techniques. In this work, the quantitative data extracted from the movies enables us to perform

statistical analysis on oxide growth rates, and such results provide additional evidence to reveal the growth mechanism.

Breakthrough scientific findings:

Our *in situ* ETEM results reveal that Cu₂O nano-islands grow in an unusual layer-by-layer growth mechanism along Cu₂O(110) planes regardless of the substrate orientation, contradicting predictions from classical theory. The statistically validated growth rate of each monolayer indicates a diffusion-limited growth mechanism that obeys the 2D island growth law in Frank-der-Merwe growth model, although the overall growth of the Cu₂O island obeys Stranski-Krastanov growth. Such layer-by-layer oxide growth is due to the more favorable Cu₂O monolayer formation caused by lower surface energy during oxidation, lower Cu and O diffusion barriers, and more favorable Cu and O adsorption energies of Cu₂O(110) over Cu₂O(100), all of which are revealed by DFT simulations. Cu detached from Cu surface steps were observed to be the source of Cu₂O island formation during the initial oxidation process, which is also at variance with classical oxidation theories.

Significance of our findings: Our findings provide a new epitaxial island growth mechanism, which will ultimately help better predict, design, and control oxide growth. In addition, the methodology of combined *in situ* TEM experiments, advanced data analysis, and theoretical simulations we presented here would significantly enhance the understanding of dynamical processes at the atomic scale.

Reviewer #3 (Remarks to the Author):

This is an interesting study of an important topic that I hope will be published in a revised form. I believe that major revisions to the current manuscript are required and give details below.

Reply: We would like to thank the reviewer for the support of our paper and providing excellent suggestions for further improving the paper.

It is certainly true, as stated by the authors, that elucidating atomic mechanisms for oxide surface growth at surfaces is of great interest. The measurements and calculations presented may well make a contribution to understanding this for Cu₂O growth on Cu. As stated below it is hard to be sure given the presentation of the data in this article. It is not, however, established clearly that the results “provide an approach to study growth dynamics of manufacturing processes” as claimed in the Abstract. Neither environmental TEM or DFT are new approaches nor is their application to the growth of oxides at surfaces. In this regard a complete article must provide at least a brief review of the previous use of ETEM, its use for similar problems and suitable citations. This is missing from the current work. For these reasons this claim should be modified or justified and a brief review provided.

The presentation of the data is couched in a description of the science that makes it very difficult to determine what the authors consider to be novel in this work and which of their conclusions are supported by the data presented.

Although environmental TEM and DFT have been used before in applications studying the growth of oxides (such as references 4, 31, 40, 43, 45 in the manuscript), these results are qualitative and only described experimentally observed phenomenon. However, in these prior studies, quantitative dynamics from *in situ* movies are not analyzed, as we have done in this submission. Such analysis is impeded by the difficulty in extracting quantitative data from *in situ* movies. In this work, as detailed in the Methods of the Supplementary Information and Supplementary Note 3, we developed an automated *in situ* TEM data analysis method that finally made this possible. After extracting the quantitative growth data from the movie, such data was analyzed in-depth via statistical techniques, further revealing the physics underlying the data. In combination with DFT simulations, this approach better evidenced the proposed mechanism. This combination of *in situ* ETEM, DFT, advanced image/movie analysis, and statistical data analysis provides a new approach to study atomic-scale dynamics at an unprecedented new level.

In order to make the novelty of our work clearer, we modified paragraph on Page 3, Line 9-14 as follows: (The modifications are in bold)

“Recent developments in *in situ* environmental transmission electron microscopy (ETEM) – with which materials systems can be examined under relevant reaction conditions – offer a solution to this problem, enabling the direct observation of growth dynamics.^{4, 29, 30} However, the observations heretofore are qualitative. Extracting statistically meaningful quantitative atomic-scale growth data from the *in situ*

movies, which is critical for understanding atomic-scale growth kinetics and mechanisms, has become the new challenge.”

The main results of the article are i) nano islands formation growing on 110 planes ii) thickness depending on the cube root of time – hence diffusion limited growth iii) a transition from step edges to substrate for the source of Cu.

Reply: We thank the reviewer for the nice summary of our results.

As the authors note, growth of Cu₂O by nano-island formation has been observed previously. Refs 25-28 in the current article for example. The authors seem to claim that the main import of this observation is that it “contradicts classical models”. As the classical models say nothing about the habit of the particles in the growth layer it isn’t clear what is meant.

Reply: We thank the reviewer for this comment. Although Cu₂O nano-island, instead of oxide film growth, has been observed previously, the growth mechanisms of such islands were not clear. Previous studies on nanoscale oxides have found striking similarities between oxide growth and heteroepitaxial thin-film growth, borrowing many concepts from thin-film growth theories to explain oxide growth (Page 2, Line 19-21). In classical models of island growth from thin-film growth theory, islands are explained to grow via the simultaneous growth of multiple layers parallel to the substrate surface (multi-layer growth mode). However, our observations suggest that the growth rate of each Cu₂O monolayer follows the diffusion-limited growth in the FvdM model, but the overall shape of the oxide follows the description of the S-K model. This is because the growth plane of the oxide is preferred along Cu₂O(110) which is not parallel to the Cu(100) substrate. Thus, we observed a new mechanism of S-K island growth, which is why we argue it “contradicts classical models”.

To make this point clearer, we have added (or modified) the following (Page 4 Line 1-14):

“3D Cu₂O islands were formed by oxidizing single-crystalline Cu films inside the ETEM at 300 °C under 0.3 Pa O₂. In agreement with previous studies^{31, 32}, these Cu₂O islands share cube-on-cube epitaxy with the Cu substrate. The Cu₂O islands on Cu(100) were reported to follow the Stranski-Krastanov growth mode, in which a transition from 2D wetting layer to 3D islands was observed beyond a critical thickness³³. According to previous models^{32, 34}, the oxide is expected to grow along the Cu surface, such as along Cu₂O(100) on Cu(100). However, as shown in Movie S1 and Fig. 1, we found the Cu₂O islands on both Cu(100) and (110) surfaces (Supporting Note 1, Figures S2-S5, and Movies S2-S4) grew along the Cu₂O(110) planes in a layer-by-layer adatom growth mode. **Such layer-by-layer growth is usually observed in Frank-van der Merwe(F-M) growth mode when the interface mismatch energy is negligible, leading to the formation of thin-film, instead of islands, along the substrate surface. Our observation shows that although the resultant Cu₂O island follows the Stranski-Krastanov growth mode, the formation of each 3D island follows layer-by-layer growth along a certain plane that is not necessarily parallel to the substrate surface, contradicting classical predictions.”**

It is also stated that the “cubic relationship in time”, in fact a cube root relationship, provides evidence of a diffusion limited growth mode that is (and here the final conclusion is unclear to me) somehow intermediate between the Stranski-Krastanov and the Frank-van der Merwe (FvdM) models. As a main conclusion of the article this is vague.

Reply: We thank the reviewer for this comment. In our manuscript, we use the terminology that “the growth of each Cu₂O monolayer obeys the **cube root relationship with time**”. This notation is consistent with classical bulk oxidate growth models (reference 35) employed in the corrosion community, which describes the cube root relationship with time as cubic in an abbreviated prose. We concur that this terminology is not consistent in other fields. To remedy this, we modified the manuscript (Page 4, Line 21) to explicitly indicate this, as:

“Despite the layer edges typically proceeding in a step-wise manner (Fig. 1g), the total projected lengths (l) all exhibited a similar smooth growth trend following a cube root relationship with time (t): $l^3 = A t$ (referred to as a cubic relationship, for short, hereafter, Figures 1h, S14, and Supporting Note 3).”

The growth rate of each Cu₂O monolayer follows the diffusion-limited growth in the FvdM model, but the overall shape of the oxide follows the description of the S-K model. This is because the growth plane of the oxide is not parallel to the substrate. So, in other words, we observed a new mechanism of S-K island growth that “contradicts classical models”. To make our conclusion clearer, we modified the conclusion section as follows:

“Our results provide direct, atomic-scale growth dynamics of 3D epitaxial oxide island growth. Instead of multilayer growth along substrate surfaces to form wedding-cake shaped islands, we found the growth of 3D epitaxial oxide islands follows a layer-by-layer growth mechanism along a preferred facet that is not parallel to the substrate surface. The growth kinetics of each oxide monolayer along the preferred facet is consistent with predictions from the diffusion-limited 2D Frank-van der Merwe growth model³⁶ for thin-films, although the overall shape of the oxide follows the Stranski-Krastanov model.”

The dependence of layer thickness on time and its relationship to the atomistic structure of the particles is potentially very interesting. The detailed discussion of this in terms of the computed DFT energetics is interesting but unconvincing.

Certainly the claim (in part c) of this discussion that the layer by layer growth of a nano-facet has a cube root time dependence because it resembles multilayer FvdM diffusion limited growth of a film is not obvious and needs to be explained more carefully.

Reply: Thanks for the suggestion. We have modified (c) to better explain the novel growth mechanism:

“c) The new Cu₂O layer grows in this adatom growth method until the edge of the layer reaches the ridge of the previous Cu₂O layer, then a new Cu₂O layer nucleates following

the steps in a) and b). **This leads to the observed layer-by-layer growth along Cu₂O(110). For each monolayer, the growth of a new Cu₂O layer on the Cu₂O(110) facet follows the diffusion-limited Frank-van der Merwe growth process with cubic growth rate $l^3 \sim t$. However, due to the preference of Cu₂O(110) over Cu₂O(100) in both kinetics and energetics, the overall Cu₂O island follows the Stranski-Krastanov growth model. When the Cu surface steps are far away, substrate Cu will feed Cu₂O growth by interfacial diffusion via place exchange with Cu vacancies in Cu₂O islands.”**

This raises the question about the role of the DFT calculations in this article. These are presented as a comprehensive examination of the atomistic growth mechanism. In reality the surface energies of facets are computed (these are available in the literature using very similar methods) and these are supplemented by the attachment of some CuxOy species and diffusion pathways for a few surface barriers. I accept that this is a very complicated process and its full elucidation in DFT is beyond current capabilities. For this reason a more balanced presentation of the results including the caveats about their application would be more convincing. If the authors believe that the surface energies, attachment mechanisms and diffusion are quantitative and in some sense representative of the overall growth they should be able to combine them to produce a kinetic growth model that correctly predicts both shape and growth rate – is that possible ?

Reply: We would like to thank the reviewer for this comment. We agree with the reviewer that the growth process is very complicated and beyond current capabilities to fully describe solely via DFT. We likewise agree that DFT simulations on their own were never meant to comprehensively represent all possible atomic diffusion events that could produce the observed experimental behaviors. Rather, these simulations were intended to supplement and validate experimental results.

In our *in situ* ETEM movies, we observed layer-by-layer Cu₂O growth along the (110) plane. From measured growth kinetics that depict longer time and larger size scale growth events than those depicted in atomistic simulations, we hypothesized that a diffusion-limited growth process was responsible for such layer-by-layer growth. However, in order to better understand the atomistic mechanisms underlying experimental observations at their earlier stages and smallest scale, DFT was necessarily applied. Previous literature, such as Bendavid *et al.* 2013 (DOI: <https://pubs.acs.org/doi/pdfplus/10.1021/jp406454c>) features simulations of surface energies of some flat Cu₂O surfaces with varied terminations and single defects, but not with matched interfacial growth of isolated Cu₂O islands on top of those surfaces. Static flat surface energetics might be adequate to predict steady-state nanoparticle shapes, such as those seen in Wulff constructions, but they cannot be applied to understand dynamic oxidation processes on their own. Our surface energy calculation results suggest that, during the oxidation process, the most favorable Cu₂O(110) configuration alternates between the two possible defect-free Cu₂O(110) terminations, while Cu₂O(100) interfacial growth requires transitioning through a very unfavorable defect-free termination. Adatom interfacial calculations further support this conclusion by demonstrating that many combinations of defects do not impact the relative favorability of the Cu₂O(110) surface over the Cu₂O(100) surface. Hence, with respect to a relatively robust set of static simulations that approach comprehensiveness, several intermediate

oxide growth states are found to bottleneck periodic layer-by-layer oxide growth along Cu₂O(100), while matching states encountered by Cu₂O(110) are thermodynamically preferred.

Besides thermodynamics, the reaction mechanisms linking these stable and transition adsorption states – which review the diffusion of Cu atoms, O atoms, and atomic clusters to form oxide island layers – are also simulated. These further evidence the relative favorability of oxide island layer growth on Cu₂O(110) surfaces over Cu₂O(100) surfaces. Such CI-NEB DFT results theoretically explain the earliest, smallest-scale diffusion and island formation events that lead to experimentally observed layer-by-layer growth along Cu₂O(110). In tandem with prior static interfacial energetics calculations and statistically verified growth dynamics that cover larger time and size scale events, such NEB results can comprehensively substantiate claims relating oxide island growth and atomic mechanistic events.

However, given the concerns addressed by this reviewer, the connections between such simulated diffusion events and statistical results do not appear to be made as clearly as intended. Much of the argumentation needed to address these concerns was already made in the Supplementary Information in sections not thoroughly detailed in the main document, particularly the “**Correlating Statistical Results with Experimental Observations**” subsection of “**Note 4: Statistical analysis of the growth rate**”.

On their own, the “surface energies, attachment mechanisms, and diffusion [barriers] are quantitative and in some sense representative of the overall [oxide] growth”, but only inasmuch as the relative presence of tested interface terminations, orientations, and related geometric features is concerned. The nucleation of new layers and growth of existing layers via interfacial diffusion are limited by Cu and O sources to expanding interfacial fronts, which were modeled in simulated structures to some degree. However, sourced Cu and O were ultimately supplied via diffusion over long atomic distances on the Cu substrates housing such growing oxides, which were not explicitly modeled in DFT due to necessitated simulation size constraints. As shown in Table S7, reviewed activation energies for horizontal diffusion on unreconstructed Cu(100) substrates are either comparable to or greater than those on observed Cu₂O(110) island surfaces for Cu (0.53 vs. 0.42 eV, respectively) and O (0.74 eV vs. 0.75 eV, respectively). Corresponding MRR Cu(100) energies are explicitly greater than their Cu₂O analogues (2.0 vs. 0.42 eV and 1.4 eV vs. 0.75 eV, respectively). Therefore, Cu/O diffusion on Cu substrates limits oxidation rates versus Cu/O diffusion across Cu₂O islands, particularly with respect to the nucleation and single layer growth of oxides on studied Cu(100) and Cu₂O(110) interfaces. Thus, numerical rates of single oxide layer nucleation and growth are determined by structures not modeled (or effectively treatable) in DFT simulations. Given these relative activation energies, the direction and arrangement of sourced Cu and O diffusing over the Cu substrate would generally determine the anisotropy or isotropy of oxide growth modes that ultimately fix oxide shape. Such diffusion is determined by the effects of reaction conditions on the Cu substrate, such as how strain introduced by adatoms on the substrate impacts their diffusion across such a substrate (DOI: 10.1016/j.commat.2014.04.0530927-0256). Thus, oxide shape cannot be directly determined from DFT simulations focusing solely on Cu₂O interfaces without their respective Cu substrates.

Nevertheless, integration of statistical conclusions with direct observation of ETEM movies – which is done in “**Correlating Statistical Results with Experimental Observations**” – can

supplement DFT simulation analysis, effectively addressing the concerns of oxide shape and growth rate entailed by proposing a kinetic model. With respect to oxide growth rates, studied concerted nucleation events (N4, Figure S16) events require Cu diffusion across Cu substrates and sequential Cu₂O layers to occur, the former of which limits Cu supply. In particular, the nucleation of a 4th oxide layer is statistically shown to be directly supplied from the 3rd and 2nd oxide layers, the growth fronts of which retreat in response to supplying Cu. Then, these fronts once again expand when Cu is supplied to them in sequence from adjacent oxide layers, forming a limiting balance of Cu supply across layers. All Cu mass transferred beyond this balance is ultimately sourced from the Cu substrate and rate limited accordingly, while the relative rates of Cu transfer across layers can be estimated using k_{sb} and k_{be} in Table S2. The combination of these diffusion rates can develop a model describing oxide growth rates both globally limited over islands (only Cu substrate diffusion) and locally limited over particular layers (Cu substrate diffusion, k_{sb} , k_{be}).

However, the computational description of Cu diffusion up and down oxide layers implies the impact of an Ehrlich-Schwöbel effect on selected oxide layers, which might render particular layers locally rate-limited by either Cu substrate diffusion or upward/downward Cu diffusion over layers. Evaluation of the particular conditions of oxide layers that select either mechanism in any case is beyond the scope of the simulation component of this work, which was used to validate the structural orientations, terminations, and related features of oxides present in experiment. Nevertheless, previous publications (for example, DOI: 10.1021/acs.jpcc.8b08944 and DOI: 10.1103/PhysRevLett.83.2608) demonstrate that the relative diffusion barriers of interfacial diffusion processes can be directly corresponded with experimental oxide island growth rates. Therefore, the ratios or differences of matched simulated diffusion barriers and oxide growth rates should be able to estimate unknown barriers or rates, and future work should be able to combine Cu substrate diffusion rates, k_{sb} , and k_{be} to construct a comprehensive kinetic model for layer-by-layer oxide growth that is also quantitatively validated by simulation.

Furthermore, oxide shape is observably controlled by concerted Cu diffusion processes, such as that described in “**Correlating Statistical Results with Experimental Observations**” (P6, Figure S12). Specifically, concerted diffusion transformed a zig-zag facet on the top layer of a Cu₂O(110) island (6th layer at that time) into a flat surface via mass transfer of Cu from adjacent layer to the topmost layer. This transformation was statistically and structurally linked to the retreat of previous adjacent oxide layer growth fronts, indicating a Cu mass balance limit between the layers limiting its oxidation. As layer-by-layer oxide growth occurs, the faceted Cu₂O island approaches becoming a flat thin film, retaining this structure as long as adjustment of the island top can occur following each new oxide layer nucleation event. However, this can only occur inasmuch as the interlayer mass balance enables Cu diffusion to the topmost layer of the island at any given time. Therefore, instantaneously observed oxide island shape is dynamically controlled by the availability or absence of Cu mass transfer to the topmost island layer, which is ultimately sourced from the Cu substrate and rate limited by Cu substrate diffusion or upward/downward Cu diffusion along the Cu₂O island. Thus, for the purposes of a kinetic model, observed oxide island shape features consistently faceted island sides or edges. Depending on the dynamics of Cu supply, oxide shape also features either a flat or faceted, Cu-deficient top that resembles a “nano-wedge” (for example, DOI: 10.1016/j.susc.2007.05.013 and DOI: 10.1016/j.jmmm.2004.09.028).

To summarily address this comment via revision to the manuscript, we have added the following passage to Page #10 (second to last paragraph of manuscript):

“Using DFT, we have investigated the energetics of several of the most probable diffusion paths, and corresponding thermodynamic states, that are proposed to underlie studied experimental observations. However, a complete understanding of studied oxide growth dynamics is beyond the capabilities of DFT alone, especially when considering the many possible concerted diffusion processes that induce concerted oxide nucleation and growth processes over multiple island layers. Comparisons between statistical conclusions and ETEM observations made in “Correlating Statistical Results with Experimental Observations” (Supplementary Note 4) demonstrate that limitations in Cu sourced from adjacent island layers contribute to observed island shapes and relative layer growth rates. Therefore, the combination of surface orientations and terminations predicted by simulation, as well as island shape and relative growth rates characterized by statistical conclusions and ETEM observations, is needed to completely depict oxide growth dynamics.”

The transition of the Cu source from step edges to substrate is an interesting result and is convincing. The videos of the oxide growth are also very interesting and a very nice example of what can be achieved with ETEM.

Reply. We thank the reviewer for noting the significance of our results.

Relatively minor point additional points are:

It is best to avoid undefined mnemonics in the Abstract

Reply: We would like to thank the reviewer for this suggestion. We have replaced TEM and DFT them back to their full names: **transmission electron microscopy and density-functional theory calculations**.

There are numerous minor errors in the grammar. These make reading the article difficult and in places obscure the meaning. I found myself rescanning sentences multiple times to extract the meaning. Proof reading or use of an accurate automatic tool should enable these to be removed and for both the clarity and readability of the article to be improved.

Reply: We would like to thank the reviewer for the suggestion. We have carefully read the manuscript and corrected several typos and improved its clarity.

REVIEWER COMMENTS

Reviewer #4 (Remarks to the Author):

Overall, the authors have successfully addressed all comments raised by Reviewer #3.

Specifically, the DFT study does provide useful insights into preferential surface energetics and diffusion mechanisms. It is well-established that simplified DFT models, while being computationally prohibitive, are not aimed at directly reproducing experimental observations faithfully, but rather should be used to compare ideal configurations that could not be easily controlled and reproduced in those experiments. In this work, it is clear that DFT simulations achieve this objective and nicely supplement the experimental data quantitatively.

As a totally optional suggestion but for completeness, in lines 239-240 of the text, the authors may consider to add Ag metal as another possible system where similar diffusion-limited layer-by-layer oxide growth mechanism along specific (111) crystallographic facets has recently been reported (Nature Communications 12, Article number: 558 (2021)).

Point-by-Point response to reviewer comments

Note: The reviewers' comments are in blue and italic, our responses are in black, and the changes made to the manuscript are highlighted in bold red.

REVIEWERS' COMMENTS

Reviewer #4 (Remarks to the Author):

Overall, the authors have successfully addressed all comments raised by Reviewer #3.

Specifically, the DFT study does provide useful insights into preferential surface energetics and diffusion mechanisms. It is well-established that simplified DFT models, while being computationally prohibitive, are not aimed at directly reproducing experimental observations faithfully, but rather should be used to compare ideal configurations that could not be easily controlled and reproduced in those experiments. In this work, it is clear that DFT simulations achieve this objective and nicely supplement the experimental data quantitatively.

As a totally optional suggestion but for completeness, in lines 239-240 of the text, the authors may consider to add Ag metal as another possible system where similar diffusion-limited layer-by-layer oxide growth mechanism along specific (111) crystallographic facets has recently been reported (Nature Communications 12, Article number: 558 (2021)).

Reply: We would like to thank the reviewer for the nice comments and suggestions on our paper. We have modified the manuscript to include this new result:

“Our findings would apply to other metals – such as Al⁴⁷, Ni-Cr^{4, 48}, Mo³⁰, Mg^{49, 50} and Ag⁵¹ – where a similar layer-by-layer oxide growth was observed for the islands, though without confirmation on the early stages.”